# High-pressure synthesis and crystal structure of iron $sp^3$-carbonate (Fe$_2$[C$_4$O$_{10}$]) featuring pyramidal [C$_4$O$_{10}$]$^{4-}$ anions

Valentin Kovalev [1] ✉, Dominik Spahr [1], Bjoern Winkler[1], Lkhamsuren Bayarjargal[1], Lena Wedek[1], Alena Aslandukova [2,3], Anna Pakhomova[4], Gaston Garbarino [4] & Elena Bykova[1]

The behavior of iron carbonates at high pressures is relevant for geological processes occurring in Earth interiors. Here, cubic iron $sp^3$-carbonate Fe$_2$[C$_4$O$_{10}$] was synthesized in diamond anvil cell by reacting Fe$_2$O$_3$ and CO$_2$ at 65(4) GPa and 3000($\pm$500) K, simulating the environment of localized thermal anomalies in the mantle. The crystal structure, determined by in situ single-crystal X-ray diffraction, features pyramidal [C$_4$O$_{10}$]$^{4-}$ anions. The experimental crystal structure corresponds to a structural model from density functional theory calculations. Experimentally determined values for zero-pressure volume $V_0$ and bulk modulus $K_0$ are: $V_0 = 1059(17)$ Å$^3$, $K_0 = 160(18)$ GPa, The DFT-calculated Raman spectrum, modeled with zinc substituting iron, matches the experimental one, supporting the structural model's accuracy. Fe$_2$[C$_4$O$_{10}$] remained stable upon decompression down to 25 GPa, below which it amorphized. DFT calculations also reveal a spin crossover of Fe$^{2+}$ cations at 95 GPa, which is significantly higher than in other Fe$^{2+}$-containing carbonates.

Carbonates are widely distributed in the geological environment. They play a significant role as a carbon source in geological processes occurring in the biosphere, hydrosphere and Earth's crust such as diagenesis, weathering and formation of ore deposits[1–5]. According to recent studies, carbonates could potentially serve as a carbon source during diamond crystallization across different geological environments, for example, as a component of carbonatites[6–10]. Moreover, carbonates are the primary constituents of marine sediments and can be transported together with water and carbonic fluids via subducting slabs into Earth's deep interior where they are involved in redox processes on the core-mantle boundary and in the formation of deep mantle melts[3,11].

So-called "conventional" carbonates, such as calcite and aragonite (Ca[CO$_3$]), dolomite (CaMg[CO$_3$]$_2$), siderite (Fe[CO$_3$]), and many other synthetic and natural species, contain planar trigonal [CO$_3$]$^{2-}$ anions, where the bonding orbitals of the carbon atoms show $sp^2$-hybridisation[12,13]. Recent experiments in diamond anvil cells (DAC) confirm that carbon follows the well-known tendency to increase the coordination number with applied pressure and that it changes its coordination from $sp^2$-trigonal to a $sp^3$-tetrahedral coordination at pressures as low as at ~20 GPa[14–16]. It has also

been shown that the tetrahedral coordination of carbon atoms can be recovered even under ambient conditions[15]. In addition, both [CO$_3$]$^{2-}$ and [CO$_4$]$^{4-}$ anions can polymerize at elevated pressures and temperatures. Such novel carbonates were obtained either by a thermal decomposition of single-source precursors[17–24] or through a chemical reaction between conventional carbonates and CO$_2$[15,16,25–28]. In case of $sp^2$-carbon, the reactions can lead to the formation of pyrocarbonates [C$_2$O$_5$]$^{2-}$ [29–32], while [CO$_4$]$^{4-}$ tetrahedra can be polymerized to four-membered pyramidal units [C$_4$O$_{10}$]$^{4-}$ [24–26], rings [C$_3$O$_9$]$^{6-}$ [19–21,33,34], four-membered linear groups [C$_4$O$_{13}$]$^{10-}$ [17,18,24] and pyroxene-like infinite chains [C$_2$O$_6$]$^{4-}$[23]. A few carbonates with isolated [CO$_4$]$^{4-}$ tetrahedra have been also reported[15,17,27,28].

The rich crystal chemistry of high-pressure $sp^3$-carbonates brings them closer to the silicates in terms of their structural diversity[35]. The degree of polymerization of the anions can be estimated using **NBO/T** (number of **n**on-**b**ridging **o**xygen per **t**etrahedron in complex anions) widely adopted for silicates[36,37]. For isolated [CO$_4$]$^{4-}$ tetrahedra the NBO/T is 4, whereas for [C$_4$O$_{10}$]$^{4-}$ pyramids the NBO/T is 1. Thus, [C$_4$O$_{10}$]$^{4-}$ units, having only one non-bridging oxygen atom per tetrahedron, are the most polymerized carbonate anions discovered so far. Only three

[1]Goethe University Frankfurt, Institute of Geosciences, 60438 Frankfurt am Main, Germany. [2]Bayerisches Geoinstitut, University of Bayreuth, 95440 Bayreuth, Germany. [3]Material Physics and Technology at Extreme Conditions, Laboratory of Crystallography, University of Bayreuth, 95440 Bayreuth, Germany. [4]European Synchrotron Radiation Facility, 38000 Grenoble, France. ✉e-mail: kovalev@kristall.uni-frankfurt.de

anhydrous carbonates with $[C_4O_{10}]^{4-}$ pyramids have been reported: isostructural $Mn_2[C_4O_{10}] - Fd\bar{3}m$[24] and $Cd_2[C_4O_{10}] - Fd\bar{3}m$[25] and $Ca_2[C_4O_{10}] - I\bar{4}2d$[26], whose crystal structure is derived from the cubic parent structures through a tetragonal distortion. However, the recent study of barium hydrogencarbonate with similar pyramids has demonstrated that $[C_4O_{10}]^{4-}$ units can also be hydrated[38]. The limited amount of experimental data on the formation conditions and properties of $[C_4O_{10}]^{4-}$ carbonates prompted us to consider the Fe-C-O system, due to the relevance of Fe-carbonates for geochemical processes such as diamond formation[39,40] and the fact that multiple Fe-carbonates, including polymerized ones, have been already found in high-pressure experiments[17,18,20,41].

Here we report a successful synthesis and structural characterization of a cubic iron $sp^3$-carbonate featuring pyramidal $[C_4O_{10}]^{4-}$ units formed in a reaction between $\eta$-$Fe_2O_3$, and $CO_2$ at 65(4) GPa. We describe its high-pressure behavior and its stability on the decompression derived both from the experimental data and from density function theory (DFT) calculations.

## Results and discussion

### Synthesis and structural characterization of $Fe_2[C_4O_{10}]$

After laser heating of $Fe_2O_3$ to ~2000 K at 65 GPa (Fig. 1a), we used XRD mapping in order to understand if a chemical reaction had occurred. We observed the appearance of new sharp reflections on the diffraction images across the heated sample. After analyzing SCXRD data collected in several points nearby the heated area, we could determine that, the peaks belong to either $\eta$-$Fe_2O_3$ with a post-perovskite structure (space group $Cmcm$, $a = 2.6752(3)$ Å, $b = 8.616(3)$ Å, $c = 6.4374(14)$ Å) or to $CO_2$-V (space group $I\bar{4}2d$, $a = 3.4862(4)$ Å, $c = 5.7085(5)$ Å) (Fig. 1b). Careful analysis of the heated area using powder XRD map data have shown that no other phases are present (Fig. 1c). The presence of $CO_2$-V corresponds to its stability field[42] and unit cell parameters are in an agreement with the previously determined values at similar pressures[43,44].

According to previous studies, $\eta$-$Fe_2O_3$ is stable in a wide pressure range from 50 to 110 GPa at temperatures above 1500 K, while at sufficiently high temperatures (>2700–3000 K), it can partially reduce forming $Fe^{2+}$-bearing oxides, such as $Fe_5O_7$[45]. The absence of $Fe_5O_7$ in the heated sample provides sufficient evidence that the temperature during heating did not exceed 2700 K.

After we found that an initial heating up to ~2000 K was insufficient to induce a reaction, we performed a second laser heating experiment at higher temperatures (~3000 K) (Fig. 2a). As a result, XRD patterns measured on a 2D grid across the sample chamber changed dramatically, suggesting a reaction between the iron oxide and $CO_2$ or carbon from diamond anvils. From the newly appeared reflections we could identify iron carbonate $Fe^{3+}_4[C_3O_{12}]$ (S.G.: $R3c$, $a = 12.9816(16)$ Å, $c = 5.3019(12)$ Å)[17] in the central part of the sample (Fig. 2c, d). This phase was previously observed among the products of thermal decomposition of siderite, $FeCO_3$, at 74(1) GPa above 1750(100) K. In contrast, on the rims of the sides of the heated sample, we have found partially reduced $Fe^{2+}$-bearing $Fe_5O_7$ (space group $C2/m$, $a = 8.8096(7)$ Å, $b = 2.6369(10)$ Å, $c = 8.0464(5)$ Å, $\beta = 105.63(8)°$)[45], suggesting that $\eta$-$Fe_2O_3$ experienced heating at temperatures above 2700 K[45] (Fig. 2c, e).

Nevertheless, several reflections could not be assigned to $Fe^{3+}_4[C_3O_{12}]$ or $Fe_5O_7$ (Fig. 3a). The unknown phase can be found in the whole sample volume (Fig. 2b, c). According to the analysis of our subsequently collected SCXRD data, these peaks belong to a novel iron carbonate with a chemical composition $Fe_2[C_4O_{10}] - Fd\bar{3}m$ ($a = 9.3992(13)$ Å). Based on charge balance considerations, this compound contains only $Fe^{2+}$. We suggest that iron from the starting $\eta$-$Fe^{3+}_2O_3$ was completely reduced after higher-temperature heating to ~3000 K at 65 GPa. The co-occurrence of $Fe_2[C_4O_{10}]$ with non-reduced $Fe^{3+}_4C_3O_{12}$ and partially reduced $Fe_5O_7$ points to the significant temperature gradients due to the non-uniform heating and insufficient thermal insulation from the diamond anvils.

The crystal structure of $Fe_2[C_4O_{10}] - Fd\bar{3}m$ at 65 GPa was refined to $R_1 = 3.31\%$, $wR_2 = 9.02\%$ with a data/parameter ratio of 126/13 (~9.6), pointing to the high quality of the refinement. For this dataset, as well as for those collected at 57, 52 and 44 GPa, the number of reflections was sufficient to refine anisotropic displacement parameters for all atoms. The resulting $R_1$ values do not exceed 5%, with data/parameter ratio from 7.2 to 8.0, indicating the good quality of the refinements.

During decompression starting at 31 GPa, a significant reduction and broadening of the reflection intensities belonging to $Fe_2[C_4O_{10}]$ (Fig. 3c), can be observed, suggesting the onset of amorphization. The number of reflections also decreases drastically. To maintain a sufficient data-to-parameter ratio (6–7 reflections per refined parameter), only the iron atoms were refined using an anisotropic approximation. The last XRD dataset for $Fe_2[C_4O_{10}]$ with reliable structure refinement ($R_1 = 7.22\%$) was collected at 25 GPa. Below this pressure, the quality of the sample deteriorated significantly, and it was no longer possible to identify the compound from the XRD pattern (Fig. 3c). The final results of the refinements of the crystal structure of $Fe_2[C_4O_{10}]$ with the atomic coordinates and bond distances at different pressure points are given in Table 1.

In the crystal structure of $Fe_2[C_4O_{10}]$ (Fig. 4a), carbon atoms are in a $sp^3$-hybridization state and coordinated by four oxygen atoms forming $[CO_4]^{4-}$ tetrahedra. Each $[CO_4]^{4-}$ polyhedron, in turn, shares their corners with three neighboring tetrahedra forming $[C_4O_{10}]^{4-}$ "super-tetrahedral" pyramidal anions (Fig. 4b). The oxygen atoms are arranged in a close cubic packing. In the super-tetrahedra bridging C-O2 bonds are longer than the terminal C-O1 bonds. At 65 GPa, C-O1 and C-O2 bonds are 1.269(6) Å and 1.3750(18) Å, respectively (Table 1). Similar differences can be observed not

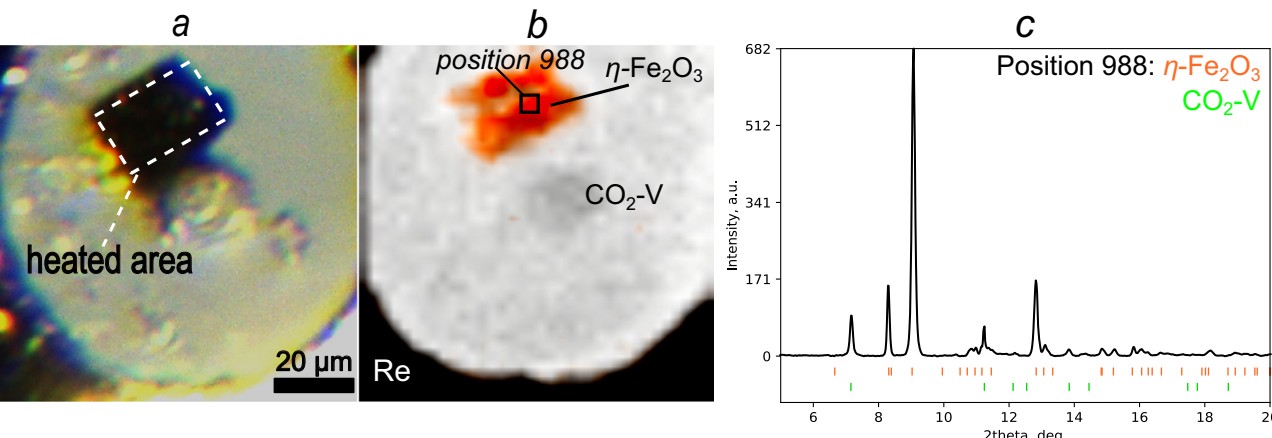

**Fig. 1 | Results of the $CO_2$-laser treatment of the sample. a** Sample chamber with $\eta$-$^{57}Fe_2O_3$ and a transparent ruby sphere in $CO_2$-pressure medium after laser heating to ~2000 K at 65 GPa; (**b**) distribution of the phases based on X-ray diffraction; (**c**) XRD pattern showing the presence of $\eta$-$Fe_2O_3$ and $CO_2$-V in the spot #988 of the XRD map.

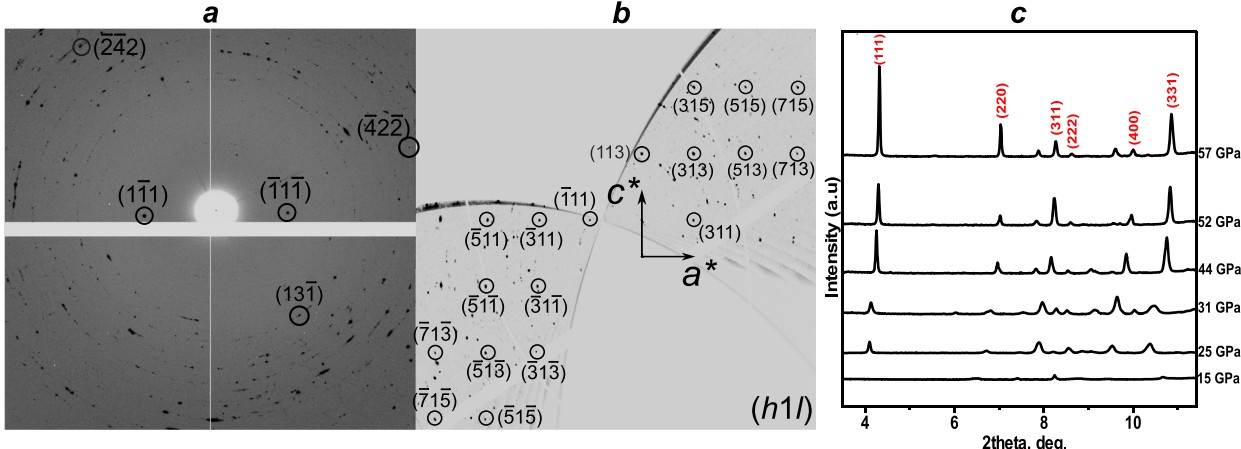

**Fig. 2 | Results of YAG-laser treatment of the sample. a** Sample chamber after heating to ~3000 K at 65 GPa localized in the central part of $\eta$-Fe$_2$O$_3$; (**b**) and (**c**)— distribution of the novel Fe$_2$[C$_4$O$_{10}$] phase across the chamber based on Raman map (band wavelength 876 cm$^{-1}$) and X-ray diffraction, respectively, together with other identified phases; XRD patterns showing identified phases in (**d**) the central part and in (**e**) the outer part of the heated sample. In the central part, the iron carbonate phase Fe$^{3+}_4$[C$_3$O$_{12}$] was detected, while in the outer part we observed partially reduced Fe$^{2+}$-bearing Fe$_5$O$_7$, indicating different temperature conditions across the sample.

**Fig. 3 | Identification of the novel phase Fe$_2$[C$_4$O$_{10}$]. a** Fragment of XRD frame at $\omega = 0°$ at 65(4) GPa with peaks belonging to Fe$_2$[C$_4$O$_{10}$] (shown by black circles). White bars are gaps in DECTRIS EIGER detector; (**b**) reciprocal space reconstruction for Fe$_2$[C$_4$O$_{10}$] corresponding to the (h1l) plane. Reflections belonging to Fe$_2$[C$_4$O$_{10}$] are marked with black circles and located at the nodes of the unwarp grid;

(**c**) XRD patterns collected for Fe$_2$[C$_4$O$_{10}$] on decompression from 57 to 15 GPa demonstrating a shift of characteristic reflections to lower-$2\theta$ region and a gradual deterioration of the sample with decreasing pressure highlighted by drops in intensities of reflections and their broadening. Characteristic reflections for Fe$_2$[C$_4$O$_{10}$] are shown by red.

**Table 1 | Details of crystal structure refinements for $Fe_2[C_4O_{10}]$ (S.G.: $Fd\overline{3}m$; Z = 8; Fe1 16 d (0, 0, 0.5), C1 32e (x, x, z), O1 32e (x, x, z), O2 48 f (x, 3/8, 3/8)) at high pressures**

| Pressure, GPa | 65(4) | 57(4) | 52(4) | 44(4) | 31(4)[a] | 25(4)[a] |
|---|---|---|---|---|---|---|
| a, Å | 9.3392(13) | 9.4330(10) | 9.4862(9) | 9.5696(7) | 9.7291(16) | 9.778(3) |
| V, Å³ | 830.4(3) | 839.4(3) | 853.6(2) | 876.36(19) | 920.9(5) | 934.9(7) |
| $\rho_{calc}$, g/cm³ | 5.115 | 5.060 | 4.976 | 4.847 | 4.612 | 4.543 |
| μ, mm⁻¹ | 1.194 | 1.517 | 1.492 | 1.453 | 1.383 | 1.362 |
| $2\Theta_{min}$ for data collection (°) | 3.948 | 4.314 | 4.290 | 4.252 | 6.832 | 4.162 |
| $2\Theta_{max}$ for data collection (°) | 39.086 | 42.318 | 41.990 | 40.404 | 36.756 | 31.140 |
| Reflections collected | 485 | 373 | 346 | 357 | 152 | 177 |
| Independent reflections | 126 | 103 | 94 | 105 | 68 | 63 |
| Independent reflections [I > 2σ(I)] | 111 | 98 | 86 | 97 | 53 | 45 |
| Refined parameters | 13 | 13 | 13 | 13 | 9 | 9 |
| $R_{int}$ | 0.0261 | 0.0274 | 0.0357 | 0.0227 | 0.0472 | 0.0693 |
| R(σ) | 0.0260 | 0.0157 | 0.0194 | 0.0127 | 0.0500 | 0.0675 |
| $R_1$ [I > 2σ(I)] | 0.0333 | 0.0331 | 0.0304 | 0.0406 | 0.0724 | 0.0722 |
| $wR_2$ [I > 2σ(I)] | 0.0908 | 0.0794 | 0.0789 | 0.1203 | 0.1815 | 0.1695 |
| $R_1$ | 0.0379 | 0.0349 | 0.0323 | 0.0422 | 0.0883 | 0.0996 |
| $wR_2$ | 0.0917 | 0.0797 | 0.0795 | 0.1211 | 0.1899 | 0.1819 |
| S (F²) | 1.161 | 1.141 | 1.124 | 1.239 | 1.192 | 1.186 |
| $\Delta\rho_{max}$ (e/Å³) | 0.65 | 0.75 | 0.60 | 0.95 | 0.72 | 1.11 |
| $\Delta\rho_{min}$ (e/Å³) | −0.68 | −0.74 | −0.67 | −0.67 | −0.68 | −0.87 |
| x, z (C1) | 0.0401(3), 0.2099(3) | 0.0404(2), 0.2096(2) | 0.0407(2), 0.2093(2) | 0.0410(4), 0.2090(4) | 0.0421(12), 0.2079(12) | 0.0417(14), 0.2083(14) |
| x, z (O1) | 0.0378(2), 0.7122(2) | 0.0376(2), 0.7124(2) | 0.0372(2), 0.7128(2) | 0.0367(3), 0.7133(3) | 0.0344(9), 0.7156(9) | 0.0349(9), 0.7151(9) |
| x (O2) | 0.0435(3) | 0.0433(2) | 0.0428(2) | 0.0416(3) | 0.0385(9) | 0.0376(11) |
| $U_{eq}$ (Fe1) | 0.0098(3) | 0.0148(3) | 0.0105(3) | 0.0177(5) | 0.0262(15) | 0.0299(17) |
| $U_{eq/iso}$ (C1) | 0.0079(5) | 0.0146(7) | 0.0092(7) | 0.0175(11) | 0.027(4) | 0.034(6) |
| $U_{eq/iso}$ (O1) | 0.0086(6) | 0.0139(6) | 0.0106(6) | 0.0176(9) | 0.028(3) | 0.024(4) |
| $U_{eq/iso}$ (O2) | 0.0081(5) | 0.0127(5) | 0.0090(5) | 0.0161(8) | 0.025(2) | 0.021(3) |
| d (Fe1…O1), Å | 2.0567(12) | 2.0656(11) | 2.0790(11) | 2.1005(17) | 2.150(6) | 2.158(6) |
| d (Fe1…O2), Å | 2.555(3) | 2.566(3) | 2.5862(15) | 2.615(3) | 2.666(8) | 2.710(8) |
| d (C1…O1), Å | 1.269(6) | 1.273(5) | 1.281(5) | 1.288(7) | 1.29(3) | 1.30(3) |
| d (C1…O2), Å | 1.3750(18) | 1.3778(17) | 1.3810(16) | 1.385(3) | 1.384(8) | 1.389(9) |
| Data collection | ESRF, ID27, EIGER2 X CdTe 9 M detector, λ ~ 0.37 Å | ESRF, ID15b, EIGER2 X CdTe 9 M detector, λ ~ 0.41 Å | | | | |
| CCDC number | 2393312 | 2393315 | 2393316 | 2393313 | 2393314 | 2393311 |

[a]Thermal displacements for C1, O1 and O2 are in isotropic approximation.

**Fig. 4 | The crystal structure of $Fe_2[C_4O_{10}]$. a** View along c; (**b**) arrangement of $[CO_4]^{4-}$ tetrahedra in a polymerized pyramidal unit $[C_4O_{10}]^{4-}$ (C-O bond lengths correspond to 65 GPa); (**c**) assemblage of $[FeO_{12}]$ icosahedra (green) and six surrounding $[C_4O_{10}]^{4-}$ pyramids (brown) through common edges.

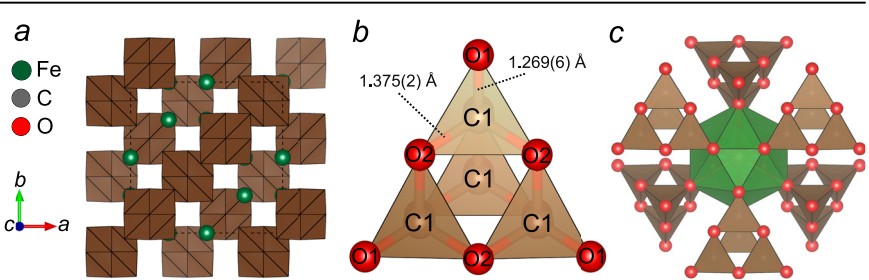

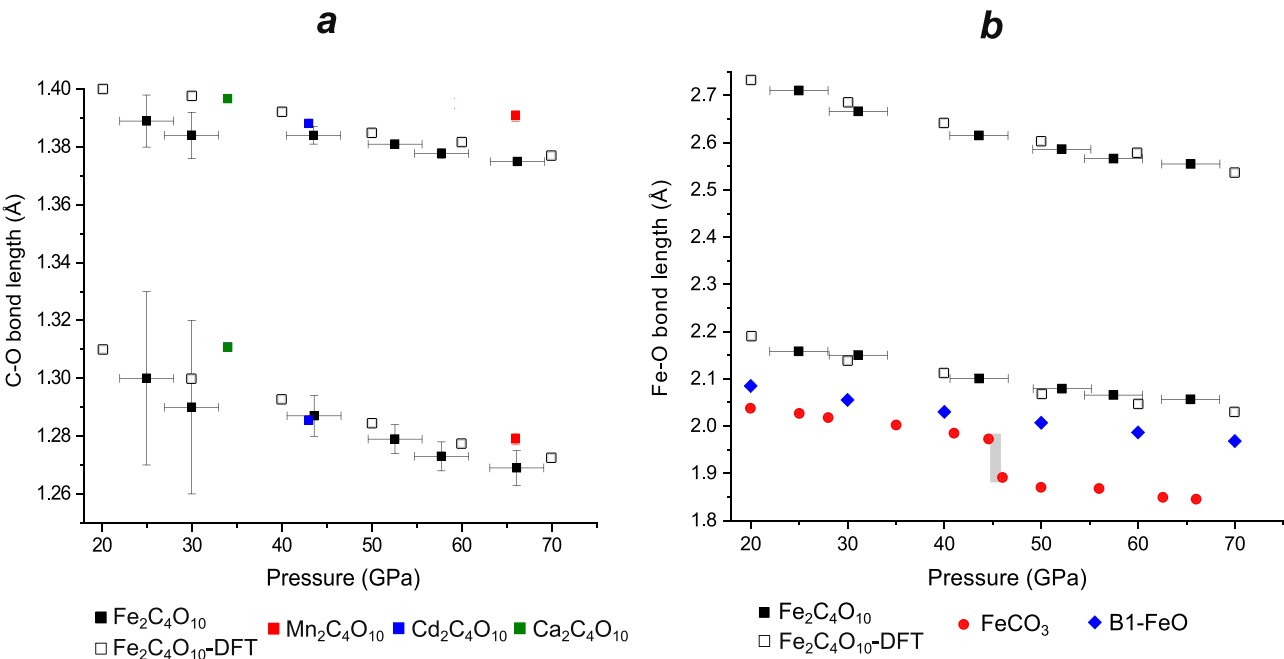

**Fig. 5 | Evolution of interatomic distances with increasing pressure. a** C-O bond lengths in high-pressure carbonates featuring $[C_4O_{10}]^{4-}$ pyramidal units as a function of pressure; (**b**) Fe-O bond lengths in $Fe_2C_4O_{10}$ and $Fe^{2+}$-bearing compounds as a function of pressure. The error bars for some data points are smaller than symbols used.

only in the pyramidal units of isostructural carbonates $Cd_2[C_4O_{10}]$[25], $Ca_2[C_4O_{10}]$[26] and $Mn_2[C_4O_{10}]$[24] (Fig. 5a), but also in other polymerized groups, such as $[C_4O_{13}]^{10-}$ truncated chains in $Fe_4[C_4O_{13}]$, $[C_3O_9]^{6-}$ three-membered rings in $Mg_3[C_3O_9]$ and $[C_2O_6]^{4-}$ pyroxene-like chains in high-pressure $Ca[CO_3]$-$P2_1/c$. At ~75 GPa, the values for terminal and bridging C-O bonds in the $[C_4O_{13}]^{10-}$ units are reported to lie within the ranges 1.275(6)-1.340(7) Å and 1.357(9)-1.394(9) Å, respectively[17]. In the case of $[C_3O_9]^{6-}$ rings in $Mg_3[C_3O_9]$ at 98 GPa, the reported bond lengths for the bridging C-O bonds are within the range 1.38(3)-1.409(19) Å, while the terminal bonds are shorter, of 1.287(18)-1.29(4) Å[46]. Similarly, for theoretically predicted $Ca[CO_3]$-$P2_1/c$ with $[C_2O_6]^{4-}$ pyroxene-like chains, terminal and bridging C-O bond lengths are reported as 1.316(0) Å and 1.415(0) Å at 60 GPa, respectively[47].

$[C_4O_{10}]^{4-}$ units form large cavities occupied by $Fe^{2+}$ cations. Each $Fe^{2+}$ cation is coordinated by twelve oxygen atoms (six Fe-O1 at a distance of 2.0567(12) Å, and six Fe-O2 at 2.555(3) Å at 65 GPa), forming a distorted icosahedron (Fig. 4c). In contrast, $Fe^{2+}$ occupies smaller sites in structures of other $Fe^{2+}$-bearing carbonates, such as $FeCO_3$ and $Fe_4C_4O_{13}$. In $Fe_4C_4O_{13}$, $Fe^{2+}$ atoms occupy bicapped prisms with a Fe-O average bond length of 2.0314 Å at 97 GPa[17], while $FeCO_3$ follows a calcite structural type, with Fe-O bond length of 1.845(3) Å at 66 GPa[48]. Based on the Fe-O bond lengths, $Fe^{2+}$ in $Fe_2[C_4O_{10}]$ remains in the high-spin state up to the highest pressure studied (65 GPa). In the calculations carried out at 50, 70, and 90 GPa, both high spin and low spin configurations were obtained. In all these calculations, the enthalpies for different high spin arrangements were quite similar, where the difference between the ferromagnetic state and a high spin antiferromagnetic state was ~0.3 eV per unit cell. The low spin configurations were substantially less stable at low pressures. However, the enthalpy difference between the low-spin and high-spin states decreased from 2.65 eV per unit cell at 50 GPa linearly to 0.36 eV per unit cell at 90 GPa. An extrapolation implies that the spin crossover would be induced at 95 GPa, which is the highest known value among known $Fe^{2+}$-bearing compounds. For example, in siderite $Fe[CO_3]$, having $Fe^{2+}$ in a smaller octahedral coordination, the spin crossover occurs between 44 and 45 GPa[48–50]. This transition pressure is well reproduced by DFT-PBE-GGA calculations such as those carried out here[51]. In wüstite, B1-FeO, with an octahedral

coordination of $Fe^{2+}$ atoms, similar to that in siderite, the spin crossover occurs at 74 GPa[52,53], but Fe-O bonds in wüstite are longer than those in siderite (Fig. 5b). Therefore, the presence of larger-volume coordination polyhedra around iron atoms can contribute to a higher pressure for spin crossover of $Fe^{2+}$ atoms in $Fe_2C_4O_{10}$.

Density functional theory (DFT) calculations for iron compounds can be demanding due to partially filled 3d-orbitals and variable spin states[54,55]. We investigated the influence of an on-site Coulomb correction with the +U-approach but found the effect to be very small. To avoid high-demand computations with complex spin systems, we performed the calculations by substituting iron with zinc atoms in the crystal structure. Zinc is slightly heavier than iron, hence we expect a minor red-shift of Raman bands; however, its ionic radius ($r_i(Zn^{2+}_{VI}) = 0.74$Å) is comparable to that of high-spin $Fe^{2+}$ ($r_i(Fe^{2+}_{VI}) = 0.78$Å)[56]. Additionally, $Zn^{2+}$ exhibits no spin alignment or spin transitions, as all 3d-orbitals are occupied. Therefore, $Zn^{2+}$ can be considered as a geometric analog for $Fe^{2+}$, making it a suitable substitute that could facilitate the performed calculations. As a result, the experimental Raman spectrum for $Fe_2[C_4O_{10}] - Fd\bar{3}m$ at 65 GPa is in a good agreement with the calculated one for hypothetical $Zn_2[C_4O_{10}] - Fd\bar{3}m$ at the same pressure (Fig. 6), which is consistent with the assumption that there is essentially no coupling of the spin arrangement with the lattice dynamics and confirming the structural model. We have also measured and calculated Raman spectra for the coexisting $CO_2$-V phase at 65 GPa and used it as a benchmark.

Three new modes at 692, 876 and 1068 $cm^{-1}$ were observed in the Raman spectrum (Fig. 6). According to group theory analysis, the crystal structure of $Fe_2[C_4O_{10}]$ has following Raman-active modes: $3A_{1g} + 3E_g + 7T_{2g}$. We could assign the observed modes as $T_{2g}$ (692 $cm^{-1}$) and $A_{1g}$ (876, 1068 $cm^{-1}$) corresponding to complex vibrations of C-O bonds in $[C_4O_{10}]^{4-}$ pyramidal anions, according to the DFT calculations.

A comparison of the experimental and DFT-calculated spectra for $Fe_2[C_4O_{10}]$ and $Zn_2[C_4O_{10}]$, respectively, demonstrates the expected consistency, even considering the complete substitution of iron with zinc in the crystal structure. Moreover, the calculated Raman spectrum for $Cd_2C_4O_{10}$[25] is also similar to that described above in terms of relative mode intensities and positions (Fig. 6).

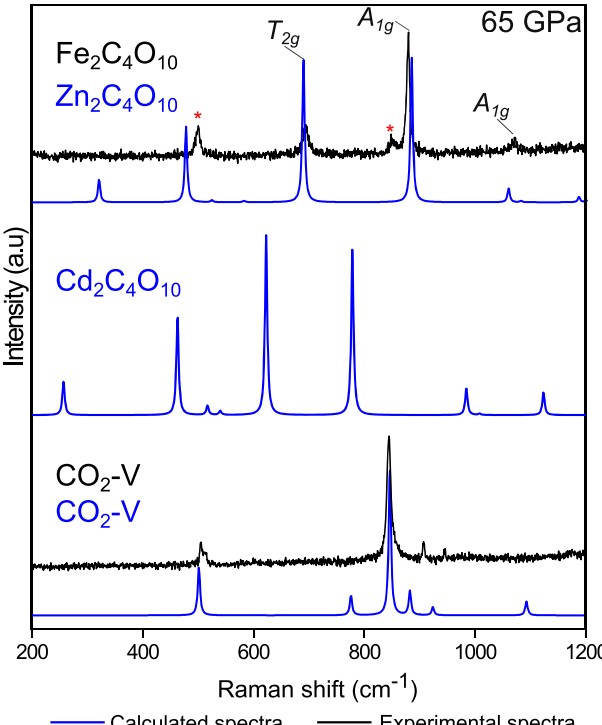

**Fig. 6 | Results of spectroscopic measurements for Fe$_2$[C$_4$O$_{10}$] and coexisting CO$_2$-V.** There is an agreement between experimental (black) Raman spectrum of Fe$_2$[C$_4$O$_{10}$] at 65 GPa and calculated one based on Zn substitution for isostructural hypothetical Zn$_2$C$_4$O$_{10}$ (blue). Characteristic modes for CO$_2$-V are marked with red asterisks (*). Calculated Raman spectrum for Zn$_2$[C$_4$O$_{10}$] also shows a striking resemblance to that of Cd$_2$[C$_4$O$_{10}$][25] confirming that these compounds are structurally similar.

## Compressional behavior of Fe$_2$[C$_4$O$_{10}$]

The compressibility of Fe$_2$[C$_4$O$_{10}$] was assessed by analyzing the unit cell volume data obtained from the solution and refinement of the crystal structure as a function of pressure[56], in our case - in a pressure range from 65 to 25 GPa. Additionally, we performed DFT-based geometrical optimizations (= relaxation) of Fe$_2$[C$_4$O$_{10}$] structures over a wider pressure range (0–90 GPa) to compare with the available experimental data. For the equation of state, the results of those calculations in which all Fe$^{2+}$ ions were in the high spin state were employed, as these were more stable than low-spin configurations up to pressures of 95 GPa.

Due to a limited number of collected pressure points and absence of an experimental $V_0$ value (Fig. 3c), we found it reasonable to fit our P-V data using the second-order Birch-Murnaghan equation of state[57] (Fig. 7a), with $V_0$ treated as a fitted parameter. For the comparison with the computed data, they were fitted using the same approach. The resulting values for bulk modulus and unit cell volume extrapolated to zero pressure were similar (Table 2). Additional verification for the choice of the equation of state (EOS) comes from the plot of normalized stress $F$ versus Eulerian strain $f$[58] (Fig. 7b). This plot shows that the strain-stress data points, within the experimental uncertainties, follow the straight horizontal line, supporting the validity of the second-order Birch-Murnaghan EOS in describing the compressibility data.

Nevertheless, the experimental data within the 25–65 GPa pressure range show a reasonable agreement with theoretical data obtained from DFT calculations (Table 2). Structural relaxation for the phase at different pressures revealed only minor differences between C-O and Fe-O bonds (Fig. 5) from experimental and calculated data. Such changes can directly influence the unit cell parameters and volume, which is obviously anticipated. Here, the discrepancy between experimental and theoretical unit cell volumes of Fe$_2$[C$_4$O$_{10}$] does not exceed 3%, pointing to an accuracy of the

structural model. Slightly higher values for the calculated unit cell volumes can be explained by the underbonding effect, where the values of bond lengths are slightly overestimated due to intrinsic short-comings of the DFT-PBE-GGA calculation approach employed here[59].

The theoretical high-pressure data for similar Ca$_2$[C$_4$O$_{10}$] and Cd$_2$[C$_4$O$_{10}$] were also analyzed here in order to reveal the effect of cation radius on the bulk modulus within this group of carbonates. The calculated datasets for these compounds were fitted using both second-order and third-order Birch-Murnaghan equations. In our considerations we used ionic radii for six-fold coordinated atoms, and in the case of iron—in a high-spin state[60]. The summarized data for different carbonates containing [C$_4$O$_{10}$]$^{4-}$ units is shown on Table 2. The bulk moduli for carbonates containing [C$_4$O$_{10}$]$^{4-}$ units follow the trend of increasing bulk modulus with decreasing cation radius. While this behavior was previously demonstrated for calcite-type carbonates[61], it is shown here for the first time in high-pressure carbonates with polymerized units. It is reasonable to expect that future synthesized carbonates with similar anions would follow the trend, highlighting the need for further investigations.

Among the iron carbonates, the compressibility and equations of state have been only analyzed for siderite Fe[CO$_3$] and Fe$_2$[CO$_3$]$_3$. It was shown that siderite exhibits an anisotropic compression behavior due to its trigonal crystal structure. The spin crossover in siderite at 44–45 GPa is accompanied by a sharp decrease of the unit cell volume from ~230 Å$^3$ (high-spin Fe[CO$_3$]) to ~ 208 Å$^3$ (low-spin Fe[CO$_3$]). There is also a significant increase of the bulk modulus from 110(2) GPa ($K'_0$ = 4.6(2)) to 148(12) GPa ($K'_0$ fixed at 5) and, consequently, in the acoustic velocity after the spin pairing is complete[48–50]. No phase and electronic transitions were identified in Fe$_2$[CO$_3$]$_3$ on compression up to 40 GPa. The bulk modulus $K_0$ derived from high-pressure XRD data is 138(34) GPa ($K'_0$ fixed at 4)[62]. In contrast to FeCO$_3$ and Fe$_2$[CO$_3$]$_3$, Fe$_2$[C$_4$O$_{10}$] contains different anionic unit, making a direct comparison of its bulk modulus and associated structural changes with those for other iron carbonates unreasonable.

## Implications to studies of high-pressure carbonates

Iron-bearing carbonates are well studied due to their significant role in deep Earth's processes, for example, formation of carbonatitic fluids and their transportation to the surface via upcoming mantle flows (plumes)[39,40]. It was previously shown that the phase with a Fe$_4$[C$_3$O$_{12}$] composition can be synthesized over a wide range of PT-conditions from different iron-bearing precursors such as FeO, Fe$_2$O$_3$ and FeOOH[41]; however, reliable structural data for this compound were initially unavailable. Later studies determined the trigonal crystal structure of Fe$_4$[C$_3$O$_{12}$] and confirmed its stability above 74(1) GPa and 1750(100) K. In addition, the crystal structure of co-existing iron carbonate Fe$_4$[C$_4$O$_{13}$] was also determined[17,63]. Our findings, based on the PT-conditions used for the synthesis of Fe$_2$[C$_4$O$_{10}$], also suggest that Fe$_2$[C$_4$O$_{10}$] could be stable under the conditions corresponding to the middle part of the lower mantle. However, the temperature used in the experiment (~3000 K) exceeds the one expected at the Earth's depth corresponding to 65 GPa (~2400–2500 K)[64]. Given the significant temperature gradients inside the sample chamber after the second heating together with experimental temperature uncertainties, we propose that localized thermal anomalies, such as those around mantle plumes, could create short-lived higher-temperature conditions (at least 200–400 K above the surrounding mantle)[65–68], where [C$_4$O$_{10}$]$^{4-}$ units could be stabilized. Such plumes could also be enriched in carbon-bearing phases[69] and hence provide a potential source of carbon for the synthesis. Consequently, the co-presence of high-pressure conditions, elevated localized temperatures, and carbon-rich environment within these thermal anomalies could create favorable conditions for the formation and stabilization of Fe$_2$[C$_4$O$_{10}$].

## Conclusions

We report the successful discovery of Fe$_2$[C$_4$O$_{10}$], a novel iron $sp^3$-carbonate with [C$_4$O$_{10}$]$^{4-}$ pyramidal groups, synthesized in diamond anvil cells under the high PT-conditions (65(4) GPa, 3000 (500) K), simulating carbon-rich environments probably existing in localized thermal anomalies in the lower

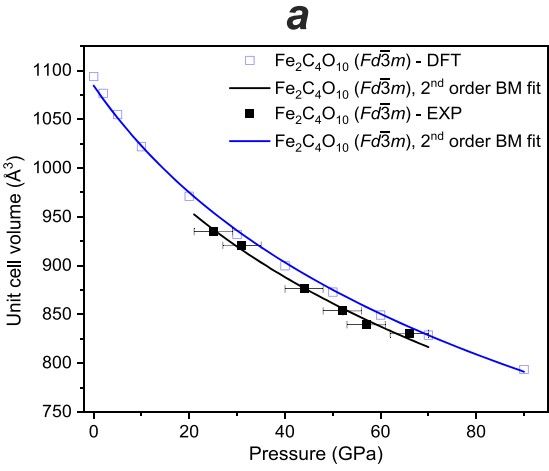

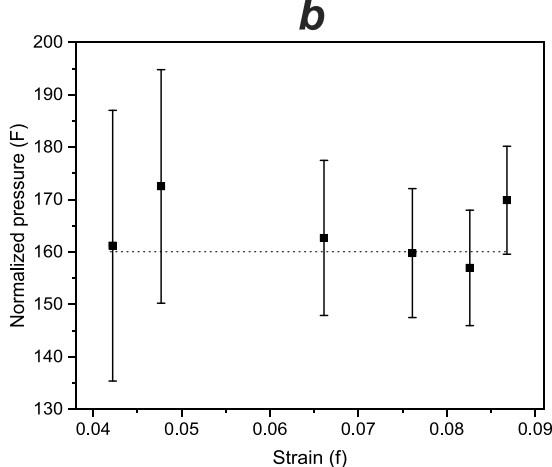

**Fig. 7 | Evolution of the unit cell volume for Fe₂[C₄O₁₀] on decompression.**
**a** Experimental and calculated unit cell volumes of $Fe_2[C_4O_{10}]$ as a function of pressure (filled and empty squares, respectively). The error bars for the volumes are smaller than symbols. The second-order Birch-Murnaghan equations of state for the calculated and experimental datasets are shown by blue and black lines, respectively; (**b**) normalized pressure ($F$) as a function of Eulerian strain ($f$) calculated for the $P$-$V$ data collected on decompression; the horizontal dash line is the second-order fit of Birch Murnaghan equation of state.

**Table 2 | Comparison of bulk moduli for M₂[C₄O₁₀] (M = Ca, Cd, Fe) as a function of ionic radius**

| Composition | Space group | Data | | $V_0$, Å³ | $K_0$, GPa | $K'_0$, GPa⁻¹ | Ionic radius, Å[a] | Ref. |
|---|---|---|---|---|---|---|---|---|
| $Fe_2[C_4O_{10}]$ | Fd3m | DFT | 2ⁿᵈ | 1084(4) | 152(3) | 4.0 | 0.78 | This work |
| | | | 3ʳᵈ | 1092(2) | 127(3) | 4.93(12) | | |
| | | EXP | 2ⁿᵈ | 1059(17) | 160(18) | 4.0 | | |
| $Cd_2[C_4O_{10}]$ | Fd3m | DFT | 2ⁿᵈ | 1186.4(6) | 132(3) | 4.0 | 0.95 | 25 |
| | | | 3ʳᵈ | 1196.8(8) | 112.2(5) | 5.21(3) | | |
| $Ca_2[C_4O_{10}]$ | I4̄2d | DFT | 2ⁿᵈ | 611(3) | 110(3) | 4.0 | 1 | 26 |
| | | | 3ʳᵈ | 616.7(2) | 93.4(4) | 4.83(2) | | |

[a]The values for ionic radii are based on[60].

mantle. The crystal structure of the compound has been determined by single-crystal X-ray diffraction, and it is similar to those of $Mn_2[C_4O_{10}]$, $Cd_2[C_4O_{10}]$ and $Ca_2[C_4O_{10}]$. These compounds, together with hydrated barium carbonate, collectively constitute a group of highly polymerized carbonates. $Fe_2[C_4O_{10}]$ shows no phase transitions within a relatively broad pressure range between 65 to 25 GPa on cold decompression. The structural and spectroscopic characteristics of $Fe_2[C_4O_{10}]$ were examined in conjunction with DFT calculations, confirming the accuracy of the structural model for $Fe_2[C_4O_{10}]$ at 65 GPa and during decompression. Additionally, DFT calculations suggest a high-spin/low-spin transformation of $Fe^{2+}$ atoms at 95 GPa.

## Methods
### Preparation of diamond anvil cell
We used DAC with the mini-BX90 design[70] to generate high pressure in the experiment. Diamond anvils with culet size of 200 μm and an opening angle of ±30° were fixed in WC seats and aligned to ensure the uniaxial compression behavior. A rhenium gasket with ~200 μm thickness was placed between diamond anvils and was pre-indented to a thickness of approximately 30 μm. A circular hole with a diameter of approximately 100 μm was drilled by a Nd:YAG pulsed laser in the center of the indentation to prepare a sample chamber.

The single crystals of $^{57}Fe$-enriched hematite (α-$^{57}Fe_2O_3$) were grown at 7 GPa and 1073 K in a 1200-tonne Sumitomo press installed at the Bavarian Geoinstitute[45]. As a precursor, a 1:1 mixture of a powder of non-enriched hematite (α-$Fe_2O_3$) of 99.998% purity and a pure powder of $^{57}Fe_2O_3$ (96.64%-enriched) was used. A particle of dark-brown α-$^{57}Fe_2O_3$ with

dimensions of $40 \times 30 \times 5$ μm was placed inside the sample chamber (Fig. 1a). Then, the DAC was cooled down to approximately 100 K, and cryogenically loaded with $CO_2$-I (dry ice)[32]. The loading process was carried out under an inert argon atmosphere to minimize contamination from atmospheric moisture. In the experiment, $CO_2$ served as both a pressure-transmitting medium and a chemical reactant[71]. Carbon dioxide is a non-hydrostatic pressure-transmitting medium, and based on previous similar studies[26,28–30], we estimate the uncertainty in the pressure determination to be around 4 GPa. Experimental pressures in the sample chamber were determined from the positions of XRD lines of $CO_2$-V[43,44]. Due to non-hydrostatic effects, the pressure was unevenly distributed across the sample chamber, therefore we determined the pressures at the specific positions where the grains of $Fe_2[C_4O_{10}]$ were located.

### Laser heating
The DAC compressed to 65 GPa was laser-heated twice. The first heating was performed using an in-house double-sided laser heating system, equipped with a pulsed $CO_2$ laser (Coherent Diamond K-250, $\lambda = 10.6$ μm)[72]. The sample was heated for 5 min with a maximum temperature of ~2000 (±100) K. The temperature was controlled by adjusting the radiation power from both sides of the sample. The laser was focused on a ~ $30 \times 30$ μm area, thereby providing uniform heating of the whole sample (Fig. 1a).

The second heating procedure was conducted using a one-sided laser-heating system at the ID27 high-pressure beamline (Nd:YAG IR laser, IPG Photonics, $\lambda = 1064$ nm, ESRF, Grenoble, France). The laser beam was focused on the central part of the sample's surface, forming a spot of

approximately $10 \times 10$ μm. The sample was heated up to a maximum temperature of ~3000 (±500) K for a few seconds (Fig. 2a). Due to the localized nature of the heating, we expect that the temperature gradients across the sample could reach ±500 K.

## X-Ray diffraction experiments and data processing

Each laser heating treatment described above was followed by a detailed X-ray diffraction mapping in order to determine the best spots for collecting of single-crystal X-ray diffraction data. Two-dimensional X-ray diffraction maps were collected from the entire sample chamber covering the sample, pressure-transmitting medium and partially rhenium gasket. The precision of the collected maps is defined by the scan ranges along the $y$ and $z$ axes and the number of steps (frames) per line. XRD maps were analyzed with the XDI software[73]; the DIOPTAS[74] software package was used for phase analysis. The calibration parameters for XDI and DIOPTAS (sample-to-detector distance, coordinates of the beam center, tilt angle, and tilt plane rotation angle of the detector images) were refined using powder XRD pattern collected from NIST $CeO_2$ standard.

X-ray diffraction data collection at 65(4) GPa was performed at ID27 high-pressure beamline at ESRF, Grenoble, France ($\lambda \sim 0.37$ Å, EIGER2 X CdTe 9 M photon-counting detector, X-ray beam size $\sim 1.5(H)$ x $1.5$ ($V$) μm²). The single-crystal XRD images were collected while rotating the DAC about a single $\omega$-axis from $-30°$ to $+30°$ with steps of 0.5° and 1 s acquisition per frame.

The DAC then was decompressed in five steps down to ~25(4) GPa and the XRD data were collected at ID15b high-pressure beamline, ESRF ($\lambda \sim 0.41$ Å, EIGER2 X CdTe 9 M photon-counting detector, X-ray beam size $\sim 1.5(H)$ x $1.5$ ($V$) μm²). Similar to the experiment at ID27, the XRD maps were used to identify the best spots for SCXRD data collection. The pressure on decompression was estimated using a diamond anvil Raman gauge[75]. Below 25 GPa, the sample became amorphous and no reasonable XRD data could be collected.

After both laser heating treatments, the reaction products were composed of multiple crystalline samples. These samples consist of randomly-oriented crystalline grains with varying sizes, resulting in a typical 2D XRD picture, where smaller grains produce powder rings, and larger grains appear as distinct spots. The DAFi[76] software, dedicated for handling multi-grain XRD data, was used to identify the most-intense crystals suitable for further processing in Crysalis^Pro (unit cell determination, integration of the reflection intensities and empirical absorption correction). The calibration of the instrument model of CrysAlis^Pro was performed using a single-crystalline vanadinite $Pb_5(VO_4)_3Cl$ ($a = 10.3174$ Å, $c = 7.3378$ Å, $Z = 2$, space group $P6_3/m$). The calibration parameters included the sample-to-detector distance, the detector's origin, offsets of the goniometer angles, rotation of the X-ray beam and the detector around the instrumental axis.

The obtained data on unit cell volumes as a function of pressure were processed using EOSFIT7-GUI software[57]. The $P$-$V$ dataset was fitted using the third-order Birch-Murnaghan equation of state[56] to derive the values for bulk modulus $K_0$, the unit cell volume extrapolated to zero pressure $V_0$ and bulk modulus derivative $K'_0$. Birch-Murnaghan equations of state are widely employed for studies of carbonates at high pressures, as well as carbonates with similar anionic groups[25,26], therefore this approach was also adopted in the current work.

## Structure solution and refinement

The crystal structures of iron oxides and carbonates obtained in this study were solved using SHELXT[77], a structure solution program that uses the algorithm of intrinsic phasing. After structure solution positions of iron atoms were determined, while the remaining atoms (carbon and oxygens) were located from the difference Fourier maps. The crystal structures were refined against $F^2$ on all data by full-matrix least-squares with the SHELXL[78] software. SHELXT and SHELXL programs were implemented in the Olex² software package[79]. The detailed summary of the crystal structure refinements at different pressures together with unit cell parameters, atomic

coordinates, and atomic thermal displacement parameters is given in Table 1.

## Raman spectroscopy

The Raman map with dimensions $60 \times 70$ μm² was recorded from the DAC at 65 GPa heated to ~3000 (±500) K. The data were collected using a dedicated setup comprised of WITec UHTS300 spectrometer (spectral resolution 0.1 cm⁻¹) with a motorized stage alpha300R confocal microscope and a DR316B low dark-current deep depletion CCD camera. Raman spectra were collected using a 532-nm excitation UHTS300S_GREEN_NIR laser source and 100 mW of its power with a 50x/0.35 Olympus SLMPL objective. The map obtained is based on 16800 Raman spectra (120 points for each of 140 map lines with 1 s integration time) collected from studied rectangular area.

## Density function theory calculations

First-principles calculations were carried out within the framework of density functional theory (DFT), employing the Perdew-Burke-Ernzerhof (PBE) exchange-correlation functional and the plane wave/pseudopotential approach implemented in the CASTEP simulation package[59,80,81]. "On the fly" norm-conserving or ultrasoft pseudopotentials generated using the descriptors in the CASTEP data base were employed in conjunction with plane waves up to a kinetic energy cutoff of 1440 eV or 630 eV, for norm-conserving and ultrasoft pseudopotentials, respectively. The accuracy of the pseudopotentials is well established[82]. Spin polarised calculations were carried out both with and without a local Coulomb correction (+U). The CASTEP implementation of DFT + U adopts a simplified, rotationally invariant approach, where only external parameter required is the effective value of the on-site Coulomb parameter, U, for each affected orbital. In the present case, a value of 2.5 eV was chosen for the Fe-$d$-orbitals. A Monkhorst-Pack grid was used for Brillouin zone integrations[83]. We used a distance between grid points of <0.023 Å⁻¹. Convergence criteria for geometry optimization included an energy change of $<5 \times 10^{-6}$ eV atom$-1$ between steps, a maximal force of <0.008 eV Å⁻¹ and a maximal component of the stress tensor <0.02 GPa. Phonon frequencies were obtained from density functional perturbation theory (DFPT) calculations[84,85]. Raman intensities were computed using DFPT with the "$2n + 1$" theorem approach[86].

## Data availability

The crystallography information files (CIFs) for the compound $Fe_2[C_4O_{10}]$ at different pressures reported in this study have been deposited at Cambridge Crystallography Data Center under deposition numbers 2393311-2393316. The data can be downloaded free of charge from https://www.ccdc.cam.ac.uk/data_request/cif.

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

## Acknowledgements

E.B. acknowledges the support of Deutsche Forschungsgemeinschaft (DFG Emmy-Noether Project No. BY101/2-1) and Johanna-Quandt-Universitäts-Stiftung. B.W. is grateful for support by the Dassault Systems Science Ambassador program and the Deutsche Forschungsgemeinschaft (DFG WI1232). We gratefully thank the European Synchrotron Radiation Facility (ESRF) for provision of synchrotron beam time under proposal numbers HC-5471 (https://doi.org/10.15151/ESRF-ES-1445120038) and CH-7016 (https://doi.org/10.15151/ESRF-ES-1719222845). We would like to thank A.P. and G.G. for assistance and support in using beamline ID27 and ID15b.

## Author contributions

V.K., D.S., L.W., A.A. and L.B. prepared and performed experiments. B.W. performed DFT calculations. A.P. and G.G. managed the synchrotron beam lines ID27 and ID15b at the ESRF. E.B. supervised the project.

## Funding

## Competing interests

The authors declare no competing interests.
