## [Peer Review file · Communications Chemistry]

High-pressure synthesis and crystal structure of iron sp^3 -carbonate ($\text{Fe}_2[\text{C}_4\text{O}_{10}]$) featuring pyramidal $[\text{C}_4\text{O}_{10}]^{4-}$ anions

Corresponding Author: Mr Valentin Kovalev

Version 0:

Reviewer comments:

Reviewer #1

(Remarks to the Author)

In the present manuscript, Kovalev et al. synthesized a new cubic iron sp^3 -carbonate $\text{Fe}_2[\text{C}_4\text{O}_{10}]$ in a laser-heated diamond anvil cell by reacting Fe_2O_3 and CO_2 at high pressure and high temperature. The crystal structure was determined by single crystal X-ray diffraction and DFT calculations. The manuscript is well-written, and the results are presented clearly. However, the authors should clarify the following issues before publication in Communications Chemistry.

(1) In line 72, the authors claimed "Based on the SCXRD," however, in the caption of Fig. 1c, the authors mentioned the "XRD powder pattern." The authors should clarify this discrepancy.

(2) In Fig. 7, the discrepancy between the theoretical and experimental equations of state (EoS) increases as the pressure decreases. Why does this occur?

(3) In Fig. 3, the diffraction patterns show some rings with spots located on them. The rings are characteristic of powder diffraction, while the spots indicate the presence of single crystals in the sample. The authors should add more details about the single-crystal diffraction analysis process.

(4) In the 2nd paragraph of page 8, the authors emphasized that Zn was used to replace Fe in $\text{Fe}_2\text{C}_4\text{O}_{10}$ for Raman simulations. However, only the Raman spectrum of $\text{Cd}_2\text{C}_4\text{O}_{10}$ was presented. I suggest ensuring consistency between the figure and the manuscript. Additionally, the authors should explain why this substitution is reasonable.

Reviewer #2

(Remarks to the Author)

Review for High-pressure synthesis and crystal structure of iron sp^3 -carbonate ($\text{Fe}_2[\text{C}_4\text{O}_{10}]$) featuring pyramidal $[\text{C}_4\text{O}_{10}]^{4-}$ anions

Abstract

The abstract is concise and informative. My only suggestion is to include the values of K and K' .

Figures

Figure 1: Please indicate pressure in the figure caption.

Figure 2: Again, please indicate pressure in the figure caption.

Figure 6: Please include symmetry/vibrational assignments for the peaks in the spectra.

Figure 7: It would be helpful to add an F - f plot to this figure (or include as a supplement). Additionally, it would be good to show an extrapolation of the BM3 EoS back to 0 GPa with a dotted line and a symbol for the experimental value of $V_0=1049.75$ from Table 2.

Manuscript text

Lines 118-119: How did you determine the onset of amorphization? It would be instructive to show any XRD patterns or Raman spectra of the sample around this pressure if they're available, or even a picture of the sample (maybe in a supplement).

Lines 209-212: The EoS discussion should be more detailed and it might be better to move lines 336-339 to this section (or at least explicitly state here that 1) V_0 was treated as a fit parameter and not based on a measurement and 2) the EoS was fit directly to the P - V data rather than F - f data). Did a fit with the Vinet EoS yield similar results? How do these results compare with the other $[\text{C}_4\text{O}_{10}]^{4-}$ carbonates?

Line 288: Since you reported the dimensions of the diamond culets, gasket, and sample chamber in micrometers you should use the same units for the dimensions of the sample. Also, since you are reporting 3 linear values (rather than a volume) you

don't need to cube the units.

Line 298: I think it's better to report something like "2000 ± 500 K" rather than "~2000 K" here.

Line 305: See comment for Line 298.

Line 328: The word "from" should be changed to "of"

Line 349: What is the resolution of the spectrometer?

General comments

As the authors point out in the introduction, the [C4O10]4- pyramidal group is interesting because it represents the most polymerized carbonate structure currently known. I think the scope of the paper could be expanded by comparing the stability and/or elasticity of the different species of [C4O10]4- carbonates. For example, you could have a figure with cation radius on the x-axis (for Ca, Cd, Mn, and Fe) and plot the pressure for the transition to the [C4O10]4- structure on the y-axis with the symbols colored by temperature. You could also try plotting bulk modulus vs. cation radius.

Reviewer #3

(Remarks to the Author)

Kovalev and co-workers present a mixed experimental/computational study on a new iron carbonate that was synthesized at high pressure in a heated diamond anvil cell. The high-pressure behavior of carbonates is of long-standing interest, due to their abundance and their importance in geological processes, and, as the authors show, the new iron carbonate can potentially be formed in Earth's mantle. The paper is potentially of great fundamental interest to the (geo)chemical community, however, the following issues need to be addressed prior to publication:

1) l. 153: It is mentioned that simulations suggest spin crossover (SCO) above 95 GPa, but no details on the calculations are given beyond the short description of the methodology at the end of the paper. On what results is this conjecture based? And, as the authors correctly point out in l. 174/5, DFT for the partially filled 3d-orbitals is "demanding", and later they speak of an "unknown spin state". DFT regularly fails to reproduce energetic spin state ordering, so without detailed presentation of the data and, most importantly, some sort of benchmark, the conjecture that SCO occurs above 95 GPa can hardly be trusted.

2) The comparison between the experimental and calculated Raman spectra does not become fully clear. In Fig. 6: Was the Raman spectrum for the Zn compound also calculated at 70 GPa? Also, what is the difference between the peaks marked by red asterisks and the black and blue lines at the very bottom that are also labeled CO2-V?

3) l. 214: The authors mention that they do not observe any pressure-induced phase transitions and that this is consistent with DFT. This is confusing. Straightforward DFT geometry optimizations at different pressures regularly fail to reproduce crystal structure transitions, therefore they cannot be used as a proof here. Or have the authors used high-pressure crystal structure prediction tools?

Version 1:

Reviewer comments:

Reviewer #1

(Remarks to the Author)

In the revised version, the authors have addressed all the reviewers' questions, and the current manuscript is now suitable for publication.

Reviewer #2

(Remarks to the Author)

The authors responded to all comments. I think the revisions have improved the manuscript.

Reviewer #3

(Remarks to the Author)

The authors have satisfactorily addressed all my comments.

Reviewer #1 (Remarks to the Author):

In the present manuscript, Kovalev et al. synthesized a new cubic iron sp^3 -carbonate $Fe_2[C_4O_{10}]$ in a laser-heated diamond anvil cell by reacting Fe_2O_3 and CO_2 at high pressure and high temperature. The crystal structure was determined by single crystal X-ray diffraction and DFT calculations. The manuscript is well-written, and the results are presented clearly. However, the authors should clarify the following issues before publication in Communications Chemistry.

Response: We thank the reviewer for the positive evaluation of our manuscript. We believe that in the revised version, we could address all reviewer's concerns and comments.

- (1) In line 72, the authors claimed “Based on the SCXRD,” however, in the caption of Fig. 1c, the authors mentioned the “XRD powder pattern.” The authors should clarify this discrepancy.
 - *Response:* We agree that the description of Fig. 1 could lead to misunderstandings regarding the experimental methods used in this study. In the first step, we indeed used multi-grain single-crystal XRD for phase identification. In the next step, we carefully examined our 2D XRD map images to verify that all diffraction reflections were accounted for by the identified phases, ensuring no unexplained peaks remained. In this context, use of powder XRD is more reliable, as it allows verification of reflections from all grains in the sample. Fig. 1c is intended to demonstrate that no additional phases, apart from n - Fe_2O_3 and CO_2 -V, are present at point #988, which is also true for other points in the heated area.
 - We added following clarifications the main text:
 - After analyzing SCXRD data collected in several points within the heated area, we could determine that, the peaks belong... Careful analysis of the heated area using powder XRD map data has shown that no other phases are present (Figure 1c).
- (2) In Fig. 7, the discrepancy between the theoretical and experimental equations of state (EoS) increases as the pressure decreases. Why does this occur?
 - *Response:* Indeed, the difference between experimental and calculated data is anticipated. Moreover, it is expected that the calculated values can be higher due to the implementation of generalized gradient approximation approach. In our case, the discrepancy between calculated and experimental data is not higher than 3%, which is reasonable.

The discrepancy between the slopes of the fitted lines was caused by an incorrect pressure estimation using the diamond anvil Raman edge. At 70 GPa, non-hydrostatic effects can result in significant pressure gradients across the sample chamber. These effects are particularly pronounced during decompression, when pressure is released unevenly across different parts of the culet. A potential solution to this issue is to determine the pressure at the specific point where the target grain of $Fe_2[C_4O_{10}]$ was located. This can be achieved, for example, by utilizing the XRD lines of the coexisting CO_2 -V phase, which has a well-characterized equation of state (EOS).

This resulted in a significant improvement in fitting EOS parameters and better correspondence with the data obtained from DFT (Figure 7a):

We added following modifications to Methods section:

Experimental pressures in the sample chamber were determined from the positions of XRD lines of $\text{CO}_2\text{-V}^{43,44}$. Due to non-hydrostatic effects, the pressure was unevenly distributed across the sample chamber, therefore we determined the pressures at the specific positions where the grains of $\text{Fe}_2[\text{C}_4\text{O}_{10}]$ were located.

- (3) In Fig. 3, the diffraction patterns show some rings with spots located on them. The rings are characteristic of powder diffraction, while the spots indicate the presence of single crystals in the sample. The authors should add more details about the single-crystal diffraction analysis process.

➤ *Response:* The laser heating in diamond anvil cells induces the formation of multi-grain samples i.e., mixtures of coarse- and fine-grain powders. Multi-grain samples consist of multiple randomly-oriented crystalline grains with varying sizes, resulting in a typical 2D XRD picture, where smaller grains produce powder rings, and larger grains appear as distinct spots.

To handle these data, we search for orientations of the most intense grains with the help of DAFi software and then process the corresponding XRD data from individual grains (unit cell determination, integration of the reflection intensities and empirical absorption correction) as if they were coming from single-crystals. The procedure is well established and described (for example, Aslandukov, A., Aslandukov, M., Dubrovinskaia, N. & Dubrovinsky, L. *Domain Auto Finder (DAFi)* program: the analysis of single-crystal X-ray diffraction data from polycrystalline samples. *J Appl Crystallogr* **55**, 1383–1391 (2022)). The details on this approach have been already given in the Methods section:

After both laser heating treatments, the reaction products were composed of multiple crystalline samples. These samples consist of randomly-oriented crystalline grains with varying sizes, resulting in a typical 2D XRD picture, where smaller grains produce powder rings, and larger grains appear as distinct spots. The DAFi⁴⁶ software dedicated for handling multi-grain XRD data was used to identify the most-intense crystals suitable for further processing in CrysAlis^{Pro} (unit cell determination, integration of the reflection intensities and empirical absorption correction).

- (4) In the 2nd paragraph of page 8, the authors emphasized that Zn was used to replace Fe in $\text{Fe}_2\text{C}_4\text{O}_{10}$ for Raman simulations. However, only the Raman spectrum of $\text{Cd}_2\text{C}_4\text{O}_{10}$ was presented. I suggest ensuring consistency between the figure and the manuscript. Additionally, the authors should explain why this substitution is reasonable.

➤ *Response:* The description for Fig.6 has been updated and $\text{Zn}_2\text{C}_4\text{O}_{10}$ is now presented here. For convenience each calculated spectrum is now blue, while the experimental ones are black. Regarding the reason of choosing Zn as a substitute atom, we hope that we gave a clear explanation in the paragraph above Fig.6. We highlighted it in the text:

Zinc is slightly heavier than iron, hence we expect a minor red-shift of Raman bands; however, its ionic radius ($r_i(\text{Zn}_{\text{VI}}^{2+}) = 0.74 \text{ \AA}$) is comparable to that of high-spin Fe^{2+} ($r_i(\text{Fe}_{\text{VI}}^{2+}) = 0.78 \text{ \AA}$)⁵⁶. Additionally, Zn^{2+} exhibits no spin alignment or spin transitions, as all 3d-orbitals are occupied. Therefore, Zn^{2+} can be considered as a geometric analog for Fe^{2+} , making it a suitable substitute that could facilitate the performed calculations.

Reviewer #2 (Remarks to the Author):

Review for High-pressure synthesis and crystal structure of iron sp³-carbonate (Fe₂[C₄O₁₀]) featuring pyramidal [C₄O₁₀]⁴⁻ anions

- Abstract

The abstract is concise and informative. My only suggestion is to include the values of K and K' .

➤ *Response:* the values for K_0 and V_0 have been added: The values for zero-pressure volume V_0 and bulk modulus K_0 from experimental data are: $V_0 = 1059(17) \text{ \AA}^3$, $K_0 = 160(18) \text{ GPa}$.

- Figures

Figure 1: Please indicate pressure in the figure caption.

➤ *Response:* the pressure value for Fig.1 has been given.

- Figure 2: Again, please indicate pressure in the figure caption.

➤ *Response:* the pressure value for Fig.2 has been given.

- Figure 6: Please include symmetry/vibrational assignments for the peaks in the spectra.

➤ *Response:* the assignment for Raman modes in spectra has been given.

- Figure 7: It would be helpful to add an F-f plot to this figure (or include as a supplement). Additionally, it would be good to show an extrapolation of the BM3 EoS back to 0 GPa with a dotted line and a symbol for the experimental value of $V_0=1049.75$ from Table 2.

➤ *Response:* We thank the reviewer for the suggestion. It should be noted that we didn't have the experimental V_0 value since the sample deteriorated under decompression below 25 GPa, and no reasonable XRD data could be obtained. In addition, after considering the comments from Reviewer #1 we found that using the diamond Raman peak for pressure estimation might be unreliable in this case due to pressure gradients propagating across the sample chamber, which are particularly pronounced during the decompression experiments. This issue created challenges in performing a reliable refinement using the 3rd order Birch-Murnaghan (BM) equation of state (EOS). In the updated version of the manuscript, we used XRD-lines of CO₂-V to determine pressure precisely at the same position where Fe₂[C₄O₁₀] was located. This adjustment allowed us to achieve better correspondence with the EOS derived from DFT calculations.

Since we had limited amount of pressure points and no experimental V_0 value, we found it reasonable to fit our data with 2nd order BM EOS. F-f plot shows that the strain-stress data, within experimental errors, follow the straight horizontal line, supporting the validity of our approach. We have added the F-f plot for experimental dataset as a Figure 7b:

Figure 7. *a*) Experimental and calculated unit cell volumes of $\text{Fe}_2[\text{C}_4\text{O}_{10}]$ as a function of pressure (filled and empty squares, respectively). The error bars for the volumes are smaller than symbols. The second-order Birch-Murnaghan equations of state for the calculated and experimental datasets are shown by blue and black lines, respectively. *b*) Normalized pressure (F) as a function of Eulerian strain (f) calculated for the P - V data collected on decompression; the horizontal dash line is the second-order fit of Birch Murnaghan equation of state.

Following lines were added to the manuscript text:

Due to a limited number of collected pressure points and absence of an experimental V_0 value (Figure 3c), we found it reasonable to fit our P - V data using the 2nd-order Birch-Murnaghan equation of state⁵⁸ (Figure 7a), with V_0 treated as a fitted parameter. For the comparison with the computed data, they were fitted using the same approach. The resulting values for bulk modulus and unit cell volume extrapolated to zero pressure were similar (Table 2). Additional verification for the choice of the equation of state (EOS) comes from the plot of normalized stress F vs. Eulerian strain f (Figure 7b) [REF to Angel 2000]. This plot shows that the strain-stress data points, within the experimental uncertainties, follow the straight horizontal line, supporting the validity of the 2nd-order Birch-Murnaghan EOS in describing the compressibility data.

- Lines 118-119: How did you determine the onset of amorphization? It would be instructive to show any XRD patterns or Raman spectra of the sample around this pressure if they're available, or even a picture of the sample (maybe in a supplement).
 - *Response:* While we provided images of the sample after the laser heating, we didn't collect photos of the sample chamber after the decompression. Nevertheless, we believe that no meaningful conclusions could be drawn from such image(s), since the synthesized carbonate was not a single crystal but rather a multi-crystalline and multi-phase, dark-colored sample.
 - The onset of amorphization was deduced from the gradual reduction in intensity and broadening of the diffraction peaks corresponding to $\text{Fe}_2\text{C}_4\text{O}_{10}$. The most pronounced changes were observed between 31 GPa and 15 GPa. At 15 GPa, no reflections corresponding to $\text{Fe}_2\text{C}_4\text{O}_{10}$ were detected in the XRD patterns. Therefore, we believe that $\text{Fe}_2\text{C}_4\text{O}_{10}$ amorphosizes below 25 GPa (which is the pressure where last SCXRD dataset with reasonable structure refinement was obtained). We have added a figure showing the evolution of $\text{Fe}_2[\text{C}_4\text{O}_{10}]$ diffraction peaks during decompression (Figure 3c):

Figure 3 c) XRD patterns collected for $\text{Fe}_2[\text{C}_4\text{O}_{10}]$ on decompression from 57 to 15 GPa demonstrating a shift of characteristic reflections to lower- 2θ region and a gradual deterioration of the sample with decreasing pressure highlighted by drops in intensities of reflections and their broadening. Characteristic reflections for $\text{Fe}_2[\text{C}_4\text{O}_{10}]$ are shown by red.

Following lines were added to the main text:

During decompression starting at 31 GPa, a significant reduction and broadening of the reflection intensities belonging to $\text{Fe}_2[\text{C}_4\text{O}_{10}]$ (Figure 3c) can be observed, suggesting the onset of amorphization. The number of reflections also decreases drastically. To maintain a sufficient data-to-parameter ratio (6-7 reflections per refined parameter), only the iron atoms were refined using an anisotropic approximation.

The last XRD dataset for $\text{Fe}_2[\text{C}_4\text{O}_{10}]$ with reliable structure refinement ($R_1 = 7.22\%$) was collected at 25 GPa. Below this pressure, the quality of the sample deteriorated significantly, and it was no longer possible to identify the compound from the XRD pattern (Figure 3c).

- Lines 209-212: The EoS discussion should be more detailed and it might be better to move lines 336-339 to this section (or at least explicitly state here that 1) V_0 was treated as a fit parameter and not based on a measurement and 2) the EoS was fit directly to the P-V data rather than F-f data).
 - *Response:* We added the details on the V_0 derived from experimental data and the type of dataset used for BM2 fits. In the manuscript we now show F-f plot, that demonstrates that the use of 2nd-order BM EOS is sufficient to describe our experimental data (Figure 7b).

Figure 7 b) Normalized pressure (F) as a function of Eulerian strain (f) calculated for the P - V data collected on decompression; the horizontal dash line is the second-order fit of Birch Murnaghan equation of state.

- Did a fit with the Vinet EoS yield similar results?
 - We attempted to fit our data with Vinet EOS, however, this resulted in poorer fits for both experimental and DFT datasets. Below, we present the fits of the DFT data using the the 2nd order Vinet (first plot) and BM EOSes (second plot):

- As a result, the 2nd order Vinet EOS significantly overestimates the bulk moduli and underestimates V_0 (DFT data):

BM2: $V_0 = 1084(4) \text{ \AA}^3$, $K_0 = 152(3) \text{ GPa}$

Vinet2: $V_0 = 1070(10) \text{ \AA}^3$, $K_0 = 235(12) \text{ GPa}$

- When fitting the data using third-order equations, the values for the bulk modulus and zero-pressure volume remain consistent. However, the value of K'_0 obtained from Vinet equation of state is significantly higher:

BM3: $V_0 = 1092(2) \text{ \AA}^3$, $K_0 = 127(3) \text{ GPa}$, $K'_0 = 4.93(1) \text{ GPa}$,

Vinet3: $V_0 = 1093(2) \text{ \AA}^3$, $K_0 = 123(3) \text{ GPa}$, $K'_0 = 5.35(11) \text{ GPa}$,

- How do these results compare with the other [C4O10]4- carbonates?
 - The calculated P-V datasets for published $\text{Ca}_2\text{C}_4\text{O}_{10}$ and $\text{Cd}_2\text{C}_4\text{O}_{10}$ were also fitted using the Birch-Murnaghan EOS. In order to properly compare the data, we fitted the datasets for Cd- and Ca-carbonates using both 2nd-order- and 3rd-order Birch-Murnaghan equations, as well as the calculated data for Fe-carbonate. The obtained values are summarized in the Table 3 (see below). Regardless of the fitting approach used, the bulk modulus increases with decreasing ionic radius of metal atom.
- Line 288: Since you reported the dimensions of the diamond culets, gasket, and sample chamber in micrometers you should use the same units for the dimensions of the sample. Also, since you are reporting 3 linear values (rather than a volume) you don't need to cube the units.

- *Response:* the correction has been given in the text. 40x30x5 μm
- Line 298: I think it's better to report something like "2000 \pm 500 K" rather than "~2000 K" here.
 - *Response:* The temperature uncertainty has been added to the description.
- Line 305: See comment for Line 298.
 - *Response:* The temperature uncertainty has been added to the description.
- Line 328: The word "from" should be changed to "of"
 - *Response:* corrected.
- Line 349: What is the resolution of the spectrometer?
 - *Response:* the spectral resolution of the spectrometer is 0.1 cm^{-1} . We added the details to the description of Methods used.
- General comments

As the authors point out in the introduction, the [C4O10]4- pyramidal group is interesting because it represents the most polymerized carbonate structure currently known. I think the scope of the paper could be expanded by comparing the stability and/or elasticity of the different species of [C4O10]4- carbonates. For example, you could have a figure with cation radius on the x-axis (for Ca, Cd, Mn, and Fe) and plot the pressure for the transition to the [C4O10]4- structure on the y-axis with the symbols colored by temperature. You could also try plotting bulk modulus vs. cation radius.

Response: We thank the reviewer for this suggestion. We used both 2nd and 3rd order Birch-Murnaghan EOS to fit P-V data of Cd₂C₄O₁₀, Ca₂C₄O₁₀ and Fe₂C₄O₁₀ in investigate the effect of cation radius on the bulk modulus. In both cases, an increase of ionic radius of the cation result in a decrease in the bulk modulus, consistent with observations made for calcite-like compounds (see Zhang, J., & Reeder, R. J. (1999). *Comparative compressibilities of calcite-structure carbonates: Deviations from empirical relations. American Mineralogist*, 84(5-6), 861-870). Due to the limited amount of the available experimental data, we found it reasonable to summarize the results in a table.

Following table was added to the manuscript text:

Table 2. Comparison of bulk moduli for M₂[C₄O₁₀] (M = Ca, Cd, Fe) as a function of ionic radius.

Composition	Space group	Data	V ₀ , Å ³	K ₀ , GPa	K' ₀ , GPa ⁻¹	Ionic radius, Å*	Ref.	
Fe ₂ [C ₄ O ₁₀]	Fd $\bar{3}m$	DFT	2 nd	1084(4)	152(3)	4.0	0.78	This work
			3 rd	1092(2)	127(3)	4.93(12)		
		EXP	2 nd	1059(17)	160(18)	4.0		
Cd ₂ [C ₄ O ₁₀]	Fd $\bar{3}m$	DFT	2 nd	1186.4(6)	132(3)	4.0	0.95	25
			3 rd	1196.8(8)	112.2(5)	5.21(3)		
Ca ₂ [C ₄ O ₁₀]	I $\bar{4}2d$	DFT	2 nd	611(3)	110(3)	4.0	1	26
			3 rd	616.7(2)	93.4(4)	4.83(2)		

*The values for ionic radii are based on⁶⁰.

Following lines were added to the manuscript text:

The theoretical high-pressure data for similar Ca₂[C₄O₁₀] and Cd₂[C₄O₁₀] compounds were also analyzed to investigate the effect of cation radius on the bulk modulus. The calculated datasets for these compounds were fitted using both second-order and third-order Birch-Murnaghan equations. In our analysis, we used ionic radii for six-fold coordinated atoms, and in the case of iron, we considered a high-spin state⁶⁰. Unfortunately, there is no bulk modulus for Mn₂[C₄O₁₀],

so it will be not further discussed. The summarized data for different carbonates containing $[C_4O_{10}]^{4-}$ units are shown in Table 3. The bulk moduli for carbonates containing $[C_4O_{10}]^{4-}$ units follow the trend of increasing bulk modulus with decreasing cation radius. While this behavior was previously demonstrated for calcite-type carbonates⁶¹, it is shown here for the first time in high-pressure carbonates with polymerized units. It is reasonable to expect that future synthesized carbonates with similar anions would follow the trend, highlighting the need for further investigations.

Reviewer #3 (Remarks to the Author):

Kovalev and co-workers present a mixed experimental/computational study on a new iron carbonate that was synthesized at high pressure in a heated diamond anvil cell. The high-pressure behavior of carbonates is of long-standing interest, due to their abundance and their importance in geological processes, and, as the authors show, the new iron carbonate can potentially be formed in Earth's mantle. The paper is potentially of great fundamental interest to the (geo)chemical community; however, the following issues need to be addressed prior to publication:

- I. 153: It is mentioned that simulations suggest spin crossover (SCO) above 95 GPa, but no details on the calculations are given beyond the short description of the methodology at the end of the paper. On what results is this conjecture based? And, as the authors correctly point out in I. 174/5, DFT for the partially filled 3d-orbitals is “demanding”, and later they speak of an “unknown spin state”. DFT regularly fails to reproduce energetic spin state ordering, so without detailed presentation of the data and, most importantly, some sort of benchmark, the conjecture that SCO occurs above 95 GPa can hardly be trusted.
 - *Response:* thanks for the remark. We rephrased the sentence in the following manner: **To avoid high-demand computations with complex spin systems, ...**
Regarding the details of our calculations, we used enthalpy minimization to derive the stability of relative stability of high-spin and low-spin states for Fe^{2+} in $Fe_2C_4O_{10}$. We hope that we clearly describe that $Fe_2C_4O_{10}$ could undergo a spin crossover above 95 GPa. However, as this is not a topic of a current research, this transition has not been further investigated. As a benchmark, we used a spin crossover in siderite $FeCO_3$ which is well-reproducible phenomenon both in theoretical and experimental research. **In the calculations carried out at 50, 70, and 90 GPa, both high spin and low spin configurations were obtained. In all these calculations, the enthalpies for different high spin arrangements were quite similar, where the difference between the ferromagnetic state and a high spin antiferromagnetic state was ~0.3 eV per unit cell. The low spin configurations were substantially less stable at low pressures. However, the enthalpy difference between the low-spin and high-spin states decreased from 2.65 eV per unit cell at 50 GPa linearly to 0.36 eV per unit cell at 90 GPa. An extrapolation implies that the spin crossover would be induced at 95 GPa, which is the highest known value among known Fe^{2+} -bearing compounds. For example, in siderite $Fe[CO_3]$, having Fe^{2+} in a smaller octahedral coordination, the spin crossover occurs between 44 and 45 GPa⁴⁷⁻⁴⁹. This transition pressure is well reproduced by DFT-PBE-GGA calculations such as those carried out here.**
- The comparison between the experimental and calculated Raman spectra does not become fully clear. In Fig. 6: Was the Raman spectrum for the Zn compound also calculated at 70 GPa? Also, what is the difference between the peaks marked by red asterisks and the black and blue lines at the very bottom that are also labeled CO₂-V?
 - *Response:* We calculated the Raman spectrum for hypothetical $Zn_2C_4O_{10}$ at the same pressure where we firstly observed $Fe_2C_4O_{10}$ in our experiment (65 GPa). In the revised version of the manuscript, we decided to use pressure values determined from the positions of XRD lines of CO₂-V, as it was shown to be more precise and consistent. The pressure value regarding all measurements and calculations is indicated in the top right corner of the Figure 6.
The calculated Raman spectrum for CO₂-V clearly reproduces the experimental one, and red asterisks above the Raman spectrum of $Fe_2C_4O_{10}$ show Raman bands belonging to CO₂-V. In other words, there are no difference between them.
- I. 214: The authors mention that they do not observe any pressure-induced phase transitions and that this is consistent with DFT. This is confusing. Straightforward DFT geometry optimizations at different pressures

regularly fail to reproduce crystal structure transitions, therefore they cannot be used as a proof here. Or have the authors used high-pressure crystal structure prediction tools?

- *Response:* Thank you for the remark. Indeed, geometrical optimization cannot give other phases in case of phase transition and cannot serve as a benchmark for these estimations. However, we used this approach to clarify stability of the current phase at different pressures (type of unit cell, unit cell parameters, C-O and Fe-O bond lengths, etc.) which also can serve as indicators on a quality of structural refinements. Also, from these calculations we can check, whether the unit cell parameters and bond lengths can be changed during the relaxation. We have not used prediction tools in this work.

We replaced this part of description: Nevertheless, the experimental data within the 25-65 GPa pressure range show a reasonable agreement with theoretical data obtained from DFT calculations (Table 3). Structural relaxations for the phase at different pressures revealed only minor differences between C-O and Fe-O bonds (Figure 5) from calculated and experimental data. Such changes can directly influence the unit cell parameters and volume, which is obviously anticipated. Here, the discrepancy between experimental and theoretical unit cell volumes of $\text{Fe}_2[\text{C}_4\text{O}_{10}]$ does not exceed 3%, pointing on an accuracy of the structural model.